# Fungal Glycoside Hydrolases Display Unique Specificities for Polysaccharides and *Staphylococcus aureus* Biofilms

**DOI:** 10.3390/microorganisms11020293

**Published:** 2023-01-23

**Authors:** Jeremy R. Ellis, James J. Bull, Paul A. Rowley

**Affiliations:** 1Department of Biological Sciences, University of Idaho, Moscow, ID 83844, USA; 2The Johns Hopkins University School of Medicine, Johns Hopkins University, Baltimore, MD 21205, USA

**Keywords:** biofilm, cellulase, amylase, xylosidase, glycoside hydrolase, *Staphylococcus aureus*

## Abstract

Commercially available cellulases and amylases can disperse the pathogenic bacteria embedded in biofilms. This suggests that polysaccharide-degrading enzymes would be useful as antibacterial therapies to aid the treatment of biofilm-associated bacteria, e.g., in chronic wounds. Using a published enzyme library, we explored the capacity of 76 diverse recombinant glycoside hydrolases to disperse *Staphylococcus aureus* biofilms. Four of the 76 recombinant glycoside hydrolases digested purified cellulose, amylose, or pectin. However, these enzymes did not disperse biofilms, indicating that anti-biofilm activity is not general to all glycoside hydrolases and that biofilm activity cannot be predicted from the activity on pure substrates. Only one of the 76 recombinant enzymes was detectably active in biofilm dispersion, an α-xylosidase from *Aspergillus nidulans*. An α-xylosidase cloned subsequently from *Aspergillus thermomutatus* likewise demonstrated antibiofilm activity, suggesting that α-xylosidases, in general, can disperse *Staphylococcus* biofilms. Surprisingly, neither of the two β-xylosidases in the library degraded biofilms. Commercial preparations of amylase and cellulase that are known to be effective in the dispersion of *Staphylococcus* biofilms were also analyzed. The commercial cellulase contained contaminating proteins with multiple enzymes exhibiting biofilm-dispersing activity. Successfully prospecting for additional antibiofilm enzymes may thus require large libraries and may benefit from purified enzymes. The complexity of biofilms and the diversity of glycoside hydrolases continue to make it difficult to predict or understand the enzymes that could have future therapeutic applications.

## 1. Introduction

Biofilms are structures formed by communities of microorganisms and enable the colonization of biotic and abiotic surfaces. Biofilms are challenging to remove by physical and chemical means, and thus pose distinct challenges to the medical and dental communities, industrial processes, and other human activities and infrastructures [1,2,3,4]. Surface colonization by bacteria is often initiated by planktonic cells adhering to the surfaces before the secretion of extracellular polymers and proteins, cementing their position. Growth of the bacterial community and the continued production of extracellular polymers result in the surface expansion and maturation of the biofilm microcolony until the disruption of the structure and the dispersal of bacteria to the environment. Dispersion can be characterized as a passive process where physical abrasion and hydrodynamic forces cause the loss of the biofilm’s integrity, and the erosion and sloughing of cellular aggregates. The active dispersion of bacterial cells is driven by the physiological state of the community and the programmed degradation of biofilm polymers by enzymes (such as proteases, glycoside hydrolases (GHs), and nucleases) and the release of planktonic cells [5,6,7,8]. Dispersed cellular aggregates and planktonic cells can seed biofilm formation in previously uncolonized environments. 

Biofilm bacteria are notoriously difficult to eradicate, and exopolysaccharide (EPS) is central to the protection of biofilm bacteria from external stressors. The composition of EPS includes—and is substantially composed of—macromolecules such as polysaccharides, proteins, nucleic acids, and even bacteriophages. Different combinations of monosaccharides define the makeup of EPS in different strains and species of bacteria. For example, homopolysaccharides such as cellulose have been identified as EPS components of biofilms [9,10]. However, most EPS are heteropolymers of modified monosaccharides, as exemplified by Psl, Pel, and alginate in the mature biofilms of *Pseudomonas aeruginosa* [11]. The diversity of EPS imparts different cohesive, adhesive, and structural properties to the biofilms and are important determinants of bacterial survival, virulence, and disease progression [3,12].

The treatment of medical biofilms is enhanced by the disruption of the biofilm’s structure. Debridement, the deliberate physical excision of biofilms at the site of infection, is commonly applied to remove portions of infected tissues [13]. Similarly, strategies must be developed to remove biofilms from medical devices [14]. There is merit in finding alternative, less invasive means of removing or disrupting the EPS. One such approach that has been attempted on many fronts is chemical: various exogenous lipids, peptides, nitric oxides, proteases, nucleases, and glycoside hydrolases (GHs) have been shown to cause the dispersion of biofilm bacteria, either by degrading the components of EPS or by triggering the intrinsic mechanisms of dispersal (reviewed in [7,8]). Enzymes known as glycoside hydrolases (GHs) have shown promise in degrading the polysaccharides in the EPS to disperse bacteria. Amylase, cellulase, dispersin B, alginate lyase, and other GHs can disperse biofilms both in vitro and in vivo [14,15,16,17,18,19,20,21,22,23,24]. Even so, the great diversity of EPS polysaccharides in any single biofilm represents a significant challenge to the therapeutic application of GHs to aid in the clearance of biofilms. This is mainly because any one enzyme is likely to degrade only a fraction of the EPS matrix and is unlikely to cause the loss of the biofilm’s integrity. However, many microbes express the enzymes needed to degrade and remodel biofilms, so there is consequently a considerable diversity of candidate GHs that may prove helpful as therapeutics, whether individually or in combination [25].

A *Pichia pastoris* library expressing a collection of recombinant GHs from filamentous fungi offers a starting point for this unique approach to bioprospecting for biofilm-dispersing enzymes. The goal of this study was to identify novel enzymes and dispersal activities [26,27]. Although commercial cellulases and amylases are effective at dispersing various biofilms in vivo, they were effective only at high concentrations, and their purities are unknown [15,17,18,28]. Here, a recombinant expression library enabled the degradation activity to be studied for a diversity of pure enzymes without the confounding effects of the impurities found in commercial enzyme mixtures. These identified specific activities are useful for identifying enzymes with greater activities than commercial preparations and could be applied to improve the treatment outcomes of chronic biofilm-associated bacterial infections.

## 2. Materials and Methods

### 2.1. Microbial Strains and Commercial Enzymes

The bacterial strain *S. aureus* SA31 was used for all biofilm experiments. *P. pastoris* strains were obtained from the Fungal Genome Stock Center (FGSC) and were maintained on yeast extract, peptone, and dextrose (YPD) media. All recombinant enzymes from the collection of Bauer et al. are listed in Appendix A [26]. Commercial amylase and cellulase were sourced from MP Biomedicals (catalog numbers 02100447-CF and 02150583-CF, respectively). Commercial pectinase was sourced from Sigma Aldrich (catalog number P2401). All commercial enzymes were dissolved in phosphate buffered saline at a pH 7.0 (PBS).

### 2.2. Cloning of an α-Xylosidase from Different Species of Fungi

The *A. thermomutatus* α-xylosidase gene was cloned using homologous recombination in the yeast *S. cerevisiae* to create the plasmid pJE001 [29]. Four PCR fragments were amplified: (1) a 2406 bp fragment containing the *A. thermomutatus* α-xylosidase 6x His-tagged gene amplified from a DNA fragment synthesized by Twist Bioscience (the primer JEx0015 had homology to Fragment #4, and JEx002 had homology to Fragment #2); (2) a 2771 bp fragment was amplified from pPICZα A using the primers JEx003 and JEx004; (3) a 2804 bp fragment containing a *URA3* gene and a 2-micron plasmid origin was amplified from the plasmid pPAR001 using the primer JEx005, which had homology to Fragment #2 and the primer JEx006; (4) a 851 bp fragment containing the portion of the pPICZα A vector, which was amplified from pPICZα using primer JEx007, which had homology to Fragment #3 and the primer JEx0016. All four fragments were used to transform the *Saccharomyces cerevisiae* strain BY4741 and were selected on a complete medium (CM) lacking uracil plates grown at 30 °C. The recovered transformants were selected on YPD-zeocin (100 μg mL^−1^) plates at 30 °C. Transformants were grown in CM lacking uracil broth overnight at 30 °C, and the plasmids were extracted. The constructed plasmid was then used to transform NEB 10-β competent *E. coli*, which were grown overnight on LB-zeocin (25 mg mL^−1^) plates. *E. coli* transformants were grown for 16 h at 30 °C in LB-zeocin (25 mg mL^−1^) broth, and the plasmid was then harvested with a Qiagen Miniprep kit. The primers JEx009 and JEx010 were used to amplify a 5885 bp region of the *E. coli* plasmid, which was then subjected to PCR cleanup. This 5885 bp PCR product was used to transform *P. pastoris* via electroporation at settings of 10 μF, 600 ohms, and 1.15 kV (Genepulser Xcell, Biorad, USA). Electroporated cells were allowed to recover in YPD broth for 9 h and then spread on YPD-zeocin (100 μg mL^−1^) plates and grown overnight at 30 °C. Selected transformant colonies were then subjected to LiOAc-SDS yeast genomic DNA extraction [30]. The genome extracts were used as a template for PCR using the primers JEx011 and JEx012 that together flank the genomically integrated *A. thermomutatus* α-xylosidase gene. The 2864 bp product was confirmed to be the *A. thermomutatus* α-xylosidase gene via Sanger sequencing. All the primers used are listed in Appendix A. The DNA sequence of pJE001 is included in Appendix A.

### 2.3. Agar Plate GH Assay

The *P. pastoris* collection was arranged in a 96-well format using the ROTOR (Singer Instruments) and plated onto BMMY agar (containing a 0.33 M potassium/sodium phosphate buffer at pH 6.0, 4.5% yeast nitrogen base with ammonium sulfate without amino acids, 1.7% methanol, and 0.00001% biotin) supplemented with carboxymethylcellulose (CMC; average molecular weight = 250,000; degree of substitution = 1.2), pectin, or starch to a final concentration of 0.9%. The plate was inverted, and the *Pichia* specimens were grown for 72 h at room temperature. Next, 200 μL of 100% methanol was added to the inverted lid at 24 and 48 h of growth. To visualize the hydrolysis zones, the CMC plates were stained for 10–15 min with 2 mg mL^−1^ of Congo Red solution and bathed for 10 min in a 1 M NaCl solution. Pectin and starch plates were stained for 3 min with 10% Lugol’s solution.

### 2.4. Expression of GH Enzymes

A single colony of each *P. pastoris* strain was inoculated into BMGY liquid media (BMMY recipe without methanol and with 3.3% glycerol) and grown overnight at 30 °C in a shaking incubator (250–300 rpm). The cells were harvested by centrifugation for 5 min at 3000× *g* (5810R, Eppendorf, Hamburg, Germany). The supernatant was decanted, and the cell pellet was inoculated into the BMMY media at OD_600_ 1.0 and placed at 30 °C with shaking (250–300 rpm). After 24 h of growth, 100% methanol was added to a final concentration of 0.5%. The culture was allowed to grow for 24 h and then centrifuged at 21,300× *g* for 2.5 min to collect the spent media supernatant containing the recombinant GHs. This was then filter sterilized in preparation for purification by affinity chromatography.

### 2.5. GH Purification (Affinity Chromatography)

Nickel affinity resin columns (HisTrap FF, Cytiva, Marlborough, UK) were equilibrated with 5 column volumes of a binding buffer (20 mM sodium phosphate, 0.5 M NaCl, and 30 mM imidazole (pH 7.4)) at a flow rate of 1 mL min^−1^ (ÄKTA Start, Cytiva, Marlborough, UK). The filter-sterilized spent BMMY *P. pastoris* expression culture was then applied to the column at a rate of 1 mL min^−1^. The column was then washed with at least 15 column volumes of the binding buffer. The recombinant GH was then eluted using a buffer of 20 mM of sodium phosphate, 0.5 M NaCl, and 500 mM imidazole (pH 7.4). The method was adapted from [26,27].

### 2.6. Mass Spectrometry

A 30 mg cellulase preparation from MP Biomedicals (catalog number 150583) was purified by size exclusion chromatography with a Tris-HCl running buffer (pH 7.4) and a HiLoad 16/600 Superdex 200 pg column (Cytiva, Marlborough, UK) attached to an NGC medium-pressure liquid chromatography system (BioRad, Hercules, CA, USA). The eluted fractions were analyzed by SDS-PAGE, excised, digested with trypsin, desalted, and then run on the Dionex LC and Orbitrap Fusion 2 for LC-MS/MS with a 30 min run time. The raw data files were analyzed using PD 2.2 and Scaffold 5. The *Aspergillus niger* reference database was combined with a list of common contaminants for the searches. Protein identification was provided by the UT Austin Center for Biomedical Research Support Biological Mass Spectrometry Facility.

### 2.7. GH Cytotoxicity Assay

An overnight *S. aureus* culture grown in TSB was 10-fold serially diluted four times across a 96-well plate. The culture was diluted in an elution buffer, 25% EtOH, or 0.5 of a mg mL^−1^ purified α-xylosidase solution to a final volume of 100 μL. *S. aureus* cells were incubated in each solution for 1 h. A 96-well pin tool was used to pin the cells onto TSB agar. The TSB agar plate was then incubated at 37 °C overnight.

### 2.8. Xylosidase Activity Assay

A 4-nitrophenyl α-D-xylopyranoside and α-xylosidase substrate from Sigma-Aldrich formed a 0.1% solution in the GH purification elution buffer. This solution was placed in a microcuvette with purified α-xylosidase to a final volume of 1 mL at enzyme concentrations of 0.25, 0.125, 0.05, and 0.005 mg mL^−1^. Fifty seconds after the purified enzyme had been added, the presence of the released o-nitrophenol was measured at 405 nm absorbance (Eppendorf 6135 BioSpectrometer). The method was adapted from [31].

### 2.9. Confirmation of the α-Xylosidase Gene

The E8 *P. pastoris* strain was subjected to genomic DNA extraction [30], and the α-xylosidase gene within the E8 strain was confirmed via diagnostic PCR. Initial denaturation for 3 min at 98 °C was followed by 30 cycles of 98 °C denaturation for 30 s, annealing at 64 °C for 30 s, and extension at 72 °C for 1 min (C-1000, BioRad, Hercules, CA, USA). The primers were GACTGCTTCTGGATGAAGTCCTACC and 5′-TCCCTCTTTCCAATCTCAAACAGC.

### 2.10. In Vitro Polystyrene Biofilm Model

An overnight *S. aureus* culture grown in TSB + 1% dextrose broth was diluted 1:100 with sterile TSB + 1% dextrose broth and subsequently dispensed into untreated sterile 96-well round-bottomed polystyrene plates (100 μL/well). The biofilms were allowed to grow for 48 h at 37 °C. Following incubation, the supernatants were aspirated, and the wells were blotted using paper towels. After this, 125 μL of the culture supernatant or the purified enzyme solution was added to each well. After treatment for one hour at room temperature, the liquid was then blotted with fresh paper towels, and the wells were stained with 0.1% crystal violet for 15 min. The excess crystal violet stain was then blotted onto paper towels, and the plate was submerged in distilled water at a 45° angle and blotted onto paper towels. After repeating the rinse and blotting step, the stained wells were dried at room temperature. To quantify the crystal violet stain, 125 μL of 30% glacial acetic acid was added to each well and left to solubilize the stain for 15 min. The liquid from each well was transferred to a clear flat-bottomed ELISA plate, and the absorbance of each well was measured at 595 nm using a 30% acetic acid blank (FLUOstar Optima, BMG Labtech, Ortenberg, Germany).

### 2.11. The In Vitro Lubbock Chronic Wound Biofilm Model

A wound-simulative biofilm model consisting of 45% Bolton’s broth, 5% laked horse blood, and 50% bovine serum was inoculated with 5 μL of an *S. aureus* culture grown overnight in 2 mL of TSB + 1% dextrose at 37 °C. The inoculated wound media were then incubated at 37 °C for 48 h. Following incubation, the residual wound media were aspirated off the resulting biofilm clot, and the biofilm clots were weighed. Treatment liquid was added (0.3–3 mL/tube), and the biofilms were placed at 37 °C for 2–4 h. Following treatment, 100 μL of the supernatant was removed, and the optical density was measured at 600 nm. An additional 100 μL of the supernatant was diluted and spread on TSB agar to measure the CFUs. The remaining supernatant was carefully aspirated, and the post-treatment biofilm clot was weighed. The biofilm clot was suspended in 1 mL of PBS, homogenized, diluted, and then spread on TSB agar for CFU enumeration. The percentage of dispersal was calculated as (CFU in the supernatant)/(CFU in the supernatant + CFU in the biofilm). The protocol was adapted from [32].

## 3. Results

### 3.1. The Degradation of Carboxymethylcellulose, Amylose, and Pectin by Fungal GHs

To discover novel biofilm-degrading GHs, we took advantage of a previously constructed library of 76 recombinant enzymes cloned from *Aspergillus nidulans*, *Aspergillus fumigatus*, and *Neurospora crassa* [26,27]. In the construction of that library, each GH gene was inserted into the genome of the yeast *Pichia pastoris* under the transcriptional control of the methanol-inducible *AOX1* promoter. Each GH was modified to include an HA epitope tag and a yeast secretion signal that enabled the extracellular export of each enzyme. As commercial preparations of cellulases and amylases are known to be capable of degrading biofilms, a library of GH-expressing *P. pastoris* cell lines was screened to hydrolyze either carboxymethylcellulose (CMC), amylose, or pectin. The library of GH-expressing *P. pastoris* strains was printed onto methanol-containing BMMY agar (to induce the expression of GHs) supplemented with each polysaccharide (Figure 1). After the strains had grown for three days, polysaccharide degradation was determined by the presence of zones of hydrolysis around each strain of *P. pastoris*. Hydrolysis was detected by staining the growth medium with Congo Red (for CMC) or Lugol’s solution (for amylose and pectin).

In all cases, the growth of *P. pastoris* on BMMY agar (with CMC, amylose, or pectin) resulted in the degradation of polysaccharides directly beneath the growing colonies (Figure 1). As evinced by the halos extending from the colonies, four GHs from *A. nidulans* hydrolyzed polysaccharides, three with cellulase activity (Figure 1, A10, C9, and D4), and one with amylase and pectinase activity (Figure 1, C8). These results were expected from previous work that identified A10, C9, and D4 as putative endo-(1,4)-glucanases that degraded CMC (NCBI accession numbers AN1285.2, AN3418.2, and AN5214.2), and C8 as a pectin methylesterase with activity against citrus pectin (NCBI accession number AN3390.2) [26]. This approach demonstrated the ability to rapidly screen GHs for degradative activity against purified polysaccharides.

### 3.2. GHs Active on Pure Polysaccharide Substrates Do Not Disperse S. aureus Biofilms

To assess the biofilm-dispersing potential of the four identified GHs with activity on CMC, amylose, and pectin, the culture supernatants harvested from each strain were tested against *S. aureus* biofilms grown on polystyrene. The supernatants were incubated with *S. aureus* biofilms and then stained with crystal violet to visualize the biofilms and measure the degree of dispersal. Commercial cellulase and amylase (from *Aspergillus niger* and *Bacillus* sp., respectively) dispersed *S. aureus* biofilms in a concentration-dependent manner (Figure 2A,B). Despite the robust hydrolytic activity of A10, C9, D4, and C8 on purified polysaccharides, the culture supernatants from these strains failed to exhibit significant *S. aureus* biofilm-dispersing activity (Appendix A). Further purification of A10, C9, and D4 confirmed that they were still able to hydrolyze CMC, amylose, and pectin (Appendix A). These results were surprising, given the robust anti-biofilm activities of commercial cellulase and amylase.

### 3.3. α-Xylosidase Disperses Biofilms

The discordance between the GHs’ activity on purified polysaccharides versus biofilm dispersion led to the screening of 76 recombinant GHs against *S. aureus* biofilms formed on polystyrene (Figure 2C). Culture supernatants were collected from all GH-expressing *P. pastoris* strains after methanol induction in liquid BMMY and were used to challenge *S. aureus* biofilms grown on microtiter plates. The initial screening of the supernatants identified a potential biofilm-degrading enzyme with more than 40% biofilm dispersal compared with the media-only control (Figure 2C, red point). After further verification, a single GH from strain E8 showed the most robust and reproducible activity against *S. aureus* biofilms on polystyrene (Tukey’s test, *p* < 0.01) (Figure 3A). A 10-fold dilution of this culture supernatant had similar biofilm dispersing activity to 0.6 mg mL^−1^ of the commercial cellulase preparation (Figure 3A). The activity of the E8 culture supernatant was also tested using Lubbock’s chronic wound biofilm model [32]. Amorphous clots formed by *S. aureus* were incubated in the E8 culture supernatant for 4 h at 37 °C. Visual inspection after treatment revealed a dramatic disaggregation of the *S. aureus*-containing clots, whereas clots incubated with the media alone remained intact (Figure 3B). The GH expressed by *P. pastoris* strain E8 was confirmed by diagnostic PCR as an α-xylosidase from the CAZY GH family 31 (NCBI accession number AN7505.2) (Appendix A).

### 3.4. Purified α-Xylosidase Also Exhibits Biofilm-Degrading Activity

To further test the biofilm-degrading activity of the E8 α-xylosidase compared with the cellulases and pectinases from strains A10, C9, D4, and C8, affinity chromatography was used to purify each GH from the culture supernatant (Appendix A). The purified E8 α-xylosidase could not degrade CMC, amylose, or pectin (Appendix A) but was able to hydrolyze PNP-α-D-xylopyranoside to *o*-nitrophenol (Appendix A). As expected, the purified E8 α-xylosidase could also effectively disperse the *S. aureus* biofilms grown on polystyrene (Figure 4A). Moreover, only 50 μg mL^−1^ of pure E8 α-xylosidase was required for significant biofilm dispersion in microtiter plates (Tukey’s test, *p* < 0.01).

Lubbock’s chronic wound model was also used to test the effectiveness of the purified α-xylosidase in disrupting a wound-like biofilm. Both α-xylosidase and commercial cellulase resulted in significant biofilm disruption compared with the buffer alone. The addition of α-xylosidase (0.17 mg mL^−1^) resulted in a 68% reduction in the weight of the biofilm, which contrasted with the 6 mg mL^−1^ required for similar results with the commercial cellulase (Figure 4B). Accompanying the reduction in the mass was a ~3-fold increase in turbidity (Figure 4C) and significant dispersal of live bacterial cells into the aqueous phase of the reaction mixture (Figure 4D). After 2 h of treatment with the buffer alone, there was a 6.8% dispersal of live bacterial cells into the solution. In contrast, the cellulase or α-xylosidase treatments resulted in 18% and 22% dispersal of live bacterial cells, respectively (Figure 4D). To determine if the α-xylosidase was toxic to bacterial cells, ~2 × 10^7^
*S. aureus* cells were serially diluted and suspended in either the buffer, α-xylosidase solution (0.5 mg mL^−1^), or ethanol (25% *v*/*v*). After two hours of incubation at room temperature, the cell suspensions were plated onto agar to measure cell viability, with ethanol causing a ~10-fold drop in viable cells. The concentration of α-xylosidase used was more than twice that required for biofilm disruption but did not result in an observable loss of bacterial viability (Figure 4E).

### 3.5. The Activities of Different Fungal Xylosidases against S. aureus Biofilms

The biofilm-disrupting activities of the E8 α-xylosidase led us to consider that other xylosidases could also be active against *S. aureus* biofilms. Two strains of *P. pastoris* expressing β-xylosidase genes from *A. nidulans* (NCBI accession numbers AN8401.2 and AN8401.2) were found to lack detectable biofilm-degrading activities; these were the only β-xylosidases in the GH enzyme library (Figure 2C). To test whether higher concentrations of β-xylosidases could disrupt biofilms, each was purified to a concentration of >1 mg mL^−1^ (Appendix A). As was shown with the cellulases A10, C9, and D4 and the pectinase C8, the purified β-xylosidase enzymes were used to challenge *S. aureus* biofilms on polystyrene but showed no dispersion (Figure 4F).

To test the generality of the biofilm-dispersing activity of fungal α-xylosidases, a homolog of the E8 α-xylosidase from CAZY Family 31 was identified in the species *Aspergillus thermomutatus* using BLASTp (NCBI accession number CDV56_102995). The putative α-xylosidase from *A. thermomutatus* (α-xylosidase*^A.therm^*) was 758 amino acids in length with 77% identity to E8 α-xylosidase. Purification of α-xylosidase*^A.therm^* produced an enzyme capable of hydrolyzing PNP-α-D-xylopyranoside (Appendix A) and was more active than E8 α-xylosidase against *S. aureus* biofilms grown on polystyrene (compare Figure 5A with Figure 4A). Lubbock’s chronic wound model was used to test whether the purified α-xylosidase*^A.therm^* could also disrupt *S. aureus* biofilms formed in blood and serum. As with the E8 xylosidase, α-xylosidase*^A.therm^* resulted in a significant >3-fold loss of biofilm mass compared with the buffer alone (Figure 5B) and a significant 5-fold increase in turbidity (Figure 5C). There was also an increase in the dispersal of live bacteria (5.27% dispersal) relative to the buffer-only control. However, it was not judged to be significantly different from the buffer-only control (Figure 5D).

### 3.6. Analysis of Commercial Cellulase Preparations Revealed a Mixture of GH Enzymes

To determine the enzymes responsible for the anti-biofilm activities of commercially prepared cellulase and amylase, their purity was assayed by SDS-PAGE. The amylase consisted of one major protein of ~55 kDa, whereas multiple proteins were identified in the cellulase preparation, with two major proteins of ~55 kDa and ~35 kDa (Figure 6A). The two major proteins in the cellulase preparation were purified by size exclusion chromatography, where each was confirmed to be monomeric, with similar molecular weights to those estimated by gel electrophoresis (Figure 6B). The putative cellulases were digested by trypsin and analyzed by MALDI-TOF mass spectrometry. This analysis identified the 65 kDa protein (Fr1) as a glucoamylase (UniProt accession number A0A117E3H6) (Appendix A) and the 36 kDa protein (Fr2) as β-xylanase (UniProt accession number A0A100I6F6) (Appendix A). The β-xylanase protein was found to have broad substrate specificity and was able to hydrolyze CMC, amylose, and pectin. In contrast, the glucoamylase was only active in the hydrolysis of amylose (Appendix A). Despite their differing activities on purified polysaccharides, both proteins were equally active in dispersing *S. aureus* biofilms (Figure 6C). When the β-xylanase (Fr2) and glucoamylase (Fr1) were mixed in a 1:1 ratio with a final protein concentration of 1 mg mL^−1^, there was an observed additive effect on biofilm dispersal (Figure 6C). Together, these data demonstrate that commercial GH preparations vary in purity and composition. Moreover, the cellulose-degrading activities of these GHs did not correlate with their ability to disperse biofilms.

## 4. Discussion and Conclusions

Bacterial biofilms present major problems in health care because they are recalcitrant to drugs even when the bacteria are sensitive to antibiotics. Physical disruption (debridement) is a common medical intervention to remove biofilms. Enzymatic debridement is another option if we can identify appropriate enzymes. Toward this end, glycoside hydrolases (GHs) are effective for the dispersion of some biofilms, most notably commercial cellulase and amylase preparations [15,16,17,18,19,20,21]. While encouraging, commercial enzyme preparations appear to be effective at high concentrations, and we have confirmed that their purity is sometimes questionable. Indeed, from work carried out here, it now seems that these preparations consist of multiple GHs with different substrate specificities. The commercial cellulase preparation analyzed in our study contained a mixture of a β-xylanase and a glucoamylase. The β-xylanase was able to hydrolyze carboxymethylcellulose (CMC), amylose, and pectin, whereas the glucoamylase was only able to hydrolyze amylose. The complexity of these preparations makes it more difficult to identify GHs that would be useful for application against biofilms.

The work described here attempted to discover novel enzymes with improved biofilm dispersal using *S. aureus* biofilms. The main points from this study were (i) the development of a protocol for purifying and testing recombinant GHs; (ii) the discovery that 99% of the diverse GHs tested were ineffective at biofilm dispersal; and (iii) that enzymatic activity on pure carbohydrate substrates did not correlate with biofilm dispersal.

We used a previously constructed collection of 76 recombinant fungal GHs expressed from the yeast *P. pastoris* [26,27]. This approach enabled us to rapidly express and purify GHs without contaminating proteins. Direct biofilm-degrading assays identified a novel α-xylosidase that was consistently effective at dispersing *S. aureus* biofilms on plastic and in a Lubbock wound model. The discovery of this enzyme validated our approach and identified a new class of enzymes that could be further applied against biofilm-associated bacterial infections. The dispersion activity appeared to be unique to α-xylosidases, as β-xylosidases were ineffective even when purified and applied at high concentrations. Previously, α-xylosidases alone have not been associated with biofilm dispersion, instead being used mainly for plant xyloglucan hydrolysis as a step in the saccharification of plant material [33,34,35,36]. Biofilm-dispersing activity was also observed in a related fungal α-xylosidase (not part of the original library), which supports the general antibiofilm activities of this type of GH. Importantly, neither of these enzymes had detectable cellulase or amylase activities.

The α-xylosidase dispersal was not predicted on the basis of the polysaccharides of bacterial EPS. The presence of xylose has been detected in biofilms formed by *P. aeruginosa* but as a minor component produced by specific strains [11]. The dispersion of *S. aureus* biofilms by α-xylosidases could indicate that xylose is an important component of their biofilms or that these α-xylosidases have additional GH activities against other polysaccharides that are integral to these biofilms. GHs can be promiscuous enzymes that often hydrolyze various substrates. Indeed, the catalytic mechanism of different GHs is often highly conserved, and substrate specificity is defined by the residues surrounding the active site. Mutations of the substrate-binding pockets can alter the specificity of GHs for different polysaccharides [37]. Alternatively, it cannot be ruled out that the interaction of α-xylosidases with the EPS triggers biofilm dispersion without carbohydrate hydrolysis. Non-enzymatic dispersal of biofilms has been observed with the addition of nitric oxide, certain fatty acids, and cyclic di-GMP [38].

Surprisingly, a few purified cellulases from the recombinant GH library could efficiently hydrolyze CMC but did not disperse *S. aureus* biofilms. This contrasts with the robust biofilm-dispersing activity of the commercial cellulase mixture, which also degraded amylose, pectin, and cellulose. Thus, the degradation of cellulose either does not contribute to the biofilm’s dispersal, or any cellulose in *S. aureus* biofilms is unavailable for degradation by cellulases in our model systems. Along the same lines, the initial agar plate screens successfully identified cellulases, amylases, and pectinases, yet these enzymes appeared largely ineffective at dispersing biofilms.

The complexity of biofilms in general is almost certainly unfathomable because of the diversity of organisms that are incorporated into these structures, and the enormous diversity of carbohydrates that are produced by bacteria and other biofilm-forming microorganisms [25]. Identifying GHs that disperse biofilms by enzymatic hydrolysis will likely require a deeper understanding of biofilm composition, especially those associated with chronic diseases. Coupled with the discovery and purification of highly active GHs that are most efficient at the disassembly of complex carbohydrates, it is feasible that optimized strategies for biofilm disruption can be developed.

## Figures and Tables

**Figure 1 microorganisms-11-00293-f001:**
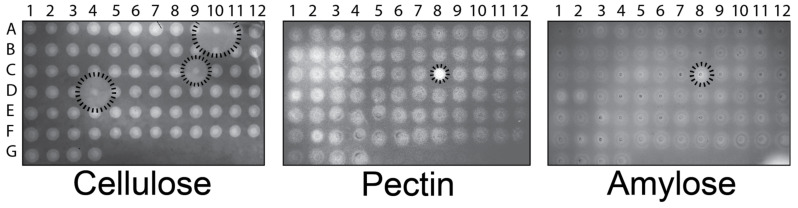
GHs expressed by *P. pastoris* can hydrolyze CMC, amylose, and pectin in agar. Recombinant *P. pastoris* strains were arranged in a 96-well plate format and printed onto BMMY agar containing methanol to induce the expression of GH. The cells were washed from the plates before staining for the presence of polysaccharides. The lack of staining indicated polysaccharide hydrolysis, as highlighted at positions A10, C8, C9, and D4.

**Figure 2 microorganisms-11-00293-f002:**
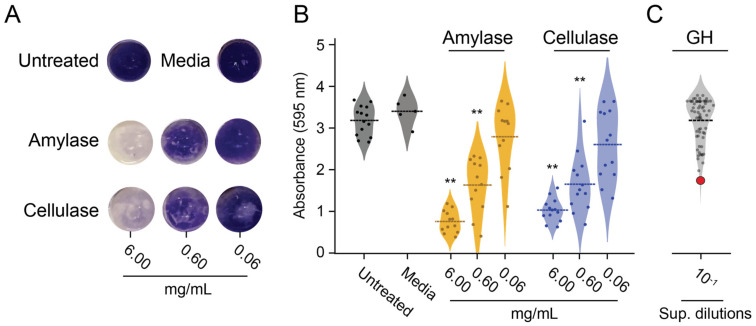
Activity of GHs in the *P. pastoris* culture supernatant against biofilms compared to commercial amylase and cellulase preparations. (**A**) Representative pictures of the dispersal of *S. aureus* biofilms from polystyrene by commercial cellulase or amylase after staining with crystal violet. (**B**) Quantification of the biofilm dispersion of commercial cellulase or amylase (6, 0.6, and 0.06 mg mL^−1^) (** *p* < 0.01, Tukey’s test). (**C**) Quantification of biofilm disruption after the addition of a 10-fold dilution of the culture supernatant (Sup. dilutions) from 76 strains of *P. pastoris* that expressed recombinant GHs. The red point represents the α-xylosidase from position E8.

**Figure 3 microorganisms-11-00293-f003:**
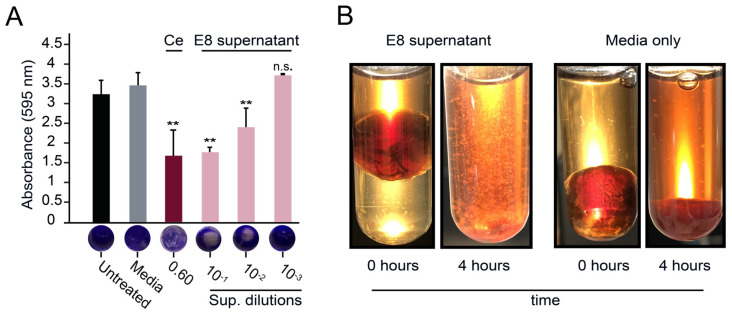
The biofilm-degrading properties of a *P. pastoris* culture supernatant containing an α-xylosidase from *A. nidulans*. (**A**) *S. aureus* biofilms were treated with spent media supernatant containing the E8 α-xylosidase and stained with crystal violet (Ce: commercial cellulase at 0.6 mg mL^−1^) (n.s. = not significant, ** *p* < 0.01, Tukey’s test). (**B**) Clots formed using the Lubbock chronic wound model were visualized before (0 min) and after (4 h) incubation in undiluted culture media with or without the E8 α-xylosidase.

**Figure 4 microorganisms-11-00293-f004:**
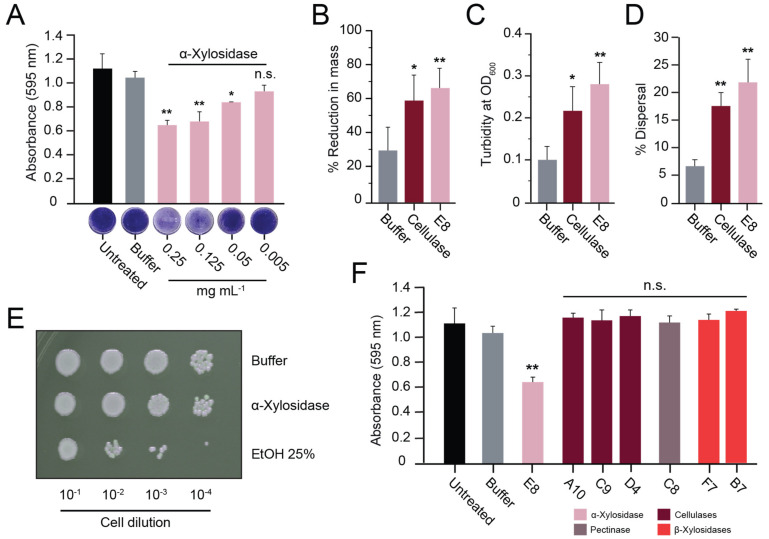
Purified α-xylosidase disperses *S. aureus* biofilms. (**A**) *S. aureus* biofilms grown on polystyrene were treated with purified α-xylosidase and stained with crystal violet. In the Lubbock chronic wound biofilm model, purified α-xylosidase and commercial cellulase were assayed for (**B**) a reduction in biofilm mass, (**C**) an increase in turbidity, and (**D**) the dispersion of viable *S. aureus* cells. (**E**) A representative image of a serial dilution of *S. aureus* grown on agar after exposure to ethanol, α-xylosidase, or the buffer only. (**F**) *S. aureus* biofilms grown on polystyrene were treated with purified GHs at concentrations of >1 mg mL^−1^ (except α-xylosidase at 0.25 mg mL^−1^). Tukey’s test was used to determine the difference from the buffer-only control (n.s. = not significant, * *p* < 0.05, ** *p* < 0.01).

**Figure 5 microorganisms-11-00293-f005:**
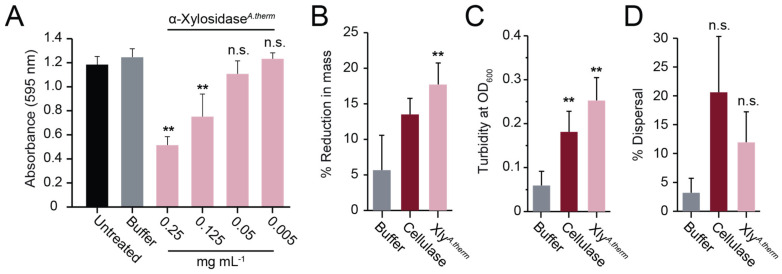
Purified α-xylosidase from *A. thermomutatus* disperses *S. aureus* biofilms. (**A**) *S. aureus* biofilms grown on polystyrene were treated with purified α-xylosidase*^A.therm^* and stained with crystal violet (*n* = 3). In the Lubbock chronic wound biofilm model, purified α-xylosidase and commercial cellulase were assayed for (**B**) a reduction in biofilm mass, (**C**) an increase in turbidity, and (**D**) the dispersion of viable *S. aureus* cells. Significance was determined relative to the buffer control (Tukey’s test, ** *p* < 0.01) (*n* = 3).

**Figure 6 microorganisms-11-00293-f006:**
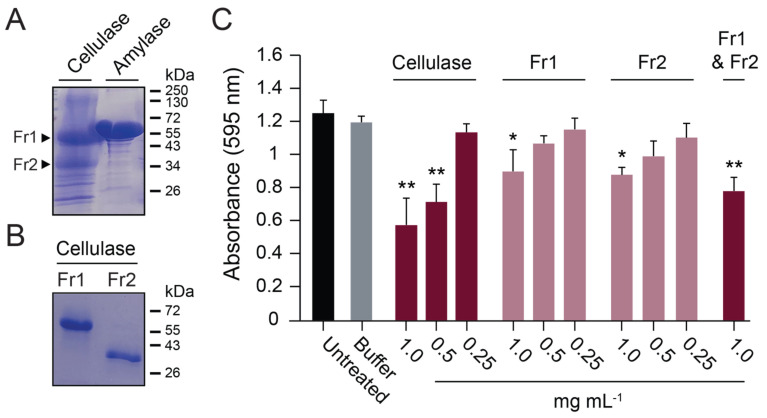
A commercial cellulase preparation contained multiple GHs that disperse biofilms. (**A**) Commercial enzyme preparations analyzed by SDS-PAGE with Coomassie blue staining. (**B**) Size exclusion chromatography purification of a glucoamylase (Fr1) and β-xylanase (Fr2) in the commercial cellulase preparation. (**C**) Fr1 and Fr2 activity against *S. aureus* biofilms grown on polystyrene. Significant differences relative to the buffer control are shown (Tukey’s test, * *p* < 0.05, ** *p* < 0.01) (*n* = 3).

## Data Availability

The data presented in this study are available in the main text and in the Appendix A.

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
