# Peer review of "Fungal Glycoside Hydrolases Display Unique Specificities for Polysaccharides and Staphylococcus aureus Biofilms"

_microorganisms, 2023, doi:10.3390/microorganisms11020293_

Round 1

Reviewer 1 Report

Based on a published enzyme library, through a series of rigorous experimental designs, the authors found that fungal glycoside hydrolases had unique specificity for polysaccharides and Staphylococcus aureus biofilms. The study is well designed and carried out. The obtained data can support the conclusion. I think that it can be accepted after addressing the following issues

1. Abstract:The significance of the research should be emphasized.

2. Line 131: The model of the centrifuge needs to be improved.

3. GH purification (affinity chromatography), Mass spectrometry and Xylosidase activity assay: References should be provided.

Author Response

  1. Abstract:The significance of the research should be emphasized.

Done. The start of the abstract now reads: “Commercially available cellulases and amylases can disperse pathogenic bacteria embedded in biofilms. This suggests that polysaccharide-degrading enzymes would be useful as antibacterial therapies to aid the treatment of biofilm-associated bacteria, i.e., in chronic wounds.”

  1. Line 131: The model of the centrifuge needs to be improved.

Done. Other instrumentation details have been added.

  1. GH purification (affinity chromatography), Mass spectrometry and Xylosidase activity assay: References should be provided.

Done. We have added a reference for the purification of the GHs and the Xylosidase assay as suggested. The mass spec analysis was done using a commercial service and the details of their protocol are provided in the methods.

Reviewer 2 Report

This study investigated unique specificities of fungal glycoside hydrolases (GHs) for polysaccharides and Staphylococcus aureus biofilms. As the authors claim, coupled with the discovery and purification of highly active GHs, it is feasible that optimized strategies for biofilm disruption can be developed. Therefore, I recommend the publication of this manuscript after following minor revisions.

1.       CMC should be defined when it first appears in the text.

2.       Degree of carboxylation of CMC should be added.

3. Information of the instruments used in this study should be added in Section 2.

Author Response

  1. CMC should be defined when it first appears in the text.

    Done

  2. Degree of carboxylation of CMC should be added.

    Done

  3. Information of the instruments used in this study should be added in Section 2.

    Done

Reviewer 3 Report

The article written by Ellis et al. deals with fungal enzymes that have potential in dispersing biofilm. The manuscript is well written and I have only a few minor comments to improve the article:

1) the Introduction should more strongly emphasize the scientific novelty of the study (i.e. in the context of e.g. https://doi.org/10.3389/fcimb.2020.00379 and clearly state the research hypothesis. Then in Conclusions there should be a reference to this hypothesis and its verification. 

2) The entire article should be more carefully reviewed from an editorial point of view, i.e. the font, style of headings, etc. should be standardized.

3) The body of the manuscript should include a table listing all the enzymes studied. 

Author Response

1) the Introduction should more strongly emphasize the scientific novelty of the study (i.e. in the context of e.g. https://doi.org/10.3389/fcimb.2020.00379 and clearly state the research hypothesis. Then in Conclusions there should be a reference to this hypothesis and its verification. 

Done: We have added some more text to more clearly outline the novelty and the goals of the study.  

Line 76: “A Pichia pastoris library expressing a collection of recombinant GHs from filamentous fungi offers a starting point for this unique approach to bioprospecting for biofilm dispersing enzymes. The goal of this study was to identify novel enzymes and dispersal activities [26,27]. Although commercial cellulases and amylases are effective at dispersing various biofilms in vivo, they were effective only at high concentrations, and their purities are unknown [15,17,18,28]. Here, a recombinant expression library enabled the degradation activity to be studied for a diversity of pure enzymes without the confounding effects of impurities found in commercial enzyme mixtures. These identified specific activities are useful for identifying enzymes with greater activities than commercial preparations that could be applied to improve the treatment outcomes of chronic biofilm-associated bacterial infections.”

Line 461 “We used a previously constructed collection of 76 recombinant fungal GHs expressed from the yeast P. pastoris [26,27]. This approach enabled us to rapidly express and purify GHs without contaminating proteins. Direct, biofilm-degrading assays identified a novel α-xylosidase that was consistently effective at dispersing S. aureus biofilms on plastic and in the semi-realistic Lubbock wound model. The discovery of this enzyme validates our approach and identified a new class of enzymes that could be further applied against biofilm-associated bacterial infection.”

2) The entire article should be more carefully reviewed from an editorial point of view, i.e. the font, style of headings, etc. should be standardized.

Will be done in the proofing stage.

3) The body of the manuscript should include a table listing all the enzymes studied. 

The enzymes are listed in the body of the text under the heading “Microbial strains and commercial enzymes.” A supplementary table of the enzymes constructer by Bauer et al. used in this study has been included.